# Professional Identity Scale for Male Nursing Students Using the Rasch Model and Latent Regression on Gender and Background Variables

**DOI:** 10.3390/healthcare10071317

**Published:** 2022-07-15

**Authors:** Renhau Li, Jiunnhorng Lou

**Affiliations:** 1Department of Psychology, Chung Shan Medical University, Taichung 40201, Taiwan; davidrhlee@yahoo.com.tw; 2Clinical Psychological Room, Chung Shan Medical University Hospital, Taichung 40201, Taiwan; 3Department of Nursing, Hsin Sheng College of Medical Care and Management, Taoyuan 325004, Taiwan

**Keywords:** male nursing students, professional identity, cognitive identity, emotional identity, behavioural identity, partial credit model, brief scale

## Abstract

This study developed a professional identity (PI) scale and compared the scores of male and female nursing students. Few studies have focused on male nursing students to understand their PI vis-à-vis related background variables relative to female ones. We recruited 384 male nursing students in Taiwan to construct the PI scale based on the Rasch model with 12 items and 3 factors, namely cognitive, emotional, and behavioural identity. The PI scale showed a good model fit in confirmatory factor analysis, with factor loadings ranging from 0.56 to 0.73. Cronbach’s alpha coefficients ranged from 0.72 to 0.79 for the three subscales and 0.88 for the entire scale. The results of the multiple latent regression analyses showed that male nursing students had higher PI in the total scale and its three factors than did female ones. Having mothers with medical or nursing-related jobs may help promote the cognitive PI of male nursing students. Experiences of caring for family members can help promote PI among female nursing students but not among male ones. Future research should focus on decreasing loss in behavioural PI for both genders after graduation and on reinforcing the association between behavioural PI and interest in nursing among male nursing students.

## 1. Introduction

The number of male nursing students is on the rise worldwide [1]. However, female nursing students continue to outnumber their male counterparts. When male students enter into a female-dominated nursing department, they are viewed as engaging in some kind of downward mobility, and their masculinity is doubted [2,3,4]. In the nursing education process, the professional identity (PI) of male nursing students is lower than that of female ones [5,6]. Individual factors contributing to the incidence of male nursing students include common reasons such as academic failure, personal/family difficulties, nursing as a wrong career choice, and financial difficulties [7,8,9]. The view that a nursing career is more suited to women is likely to act as a barrier to men who are interested in pursuing the career track [10]. Therefore, educators should give male nursing students with lower PI more targeted educational support, because PI is an integral aspect of being a professional and how individuals view and present themselves [11]. Professional socialisation in nursing education and clinical placements help develop PI [12,13]. The social learning theory, communities of practice, could be capable of providing a theoretical base for programs designed to support professional identity formation [14].

PI is a person’s perception of themselves within a profession or the collective identity of the profession [7]. The development of PI is a continuous process that is influenced by several factors including practical and professional socialisation experience [15]. PI is one of the main outcomes of education and socialisation in academic disciplines. It is also an effective factor in ensuring employee commitment [16]. The development of PI begins during nursing education. Male nursing students are the future of the nursing workforce. PI is a dynamic phenomenon that continues to evolve from university study into a health professionals’ work life [17]. However, research has shown that after graduating, most male nurses leave nursing within four years of starting their careers [1,18,19]. The PI of male nursing students was lower than that of their female counterparts [10,20]. PI influences the willingness of male students to enter into the nursing profession [8,21,22]. If men enter the nursing profession at the same rate as women, there will be no shortage of nurses [23,24].

PI is an important and universal concept in the field of nursing as it affects male nurses’ perceptions of their roles in nursing and their retention rates [25,26]. It serves as an index to predict an individual’s entry into the nursing profession [27]. PI is an individual’s view of the goal, social value, and other factors of the occupation, which is consistent with the social evaluation and expectation of the occupation [26,28,29]. Male nursing students may think that nursing lacks a sense of pride and happiness, and a few of them may choose to change careers after graduation. They tend to think that nursing cannot meet the needs of a man vis-à-vis respect and self-realisation [30,31,32]. Some studies have indicated that gender stereotypes operate as negative factors influencing the PI of male nursing students [8,33].

Compared to female nurses, turnover rates are higher and retention times are shorter among male ones [34,35,36]. College education is critical to PI formation among male nursing students; if they are educated to have a higher sense of nursing PI, they can also have a better nursing career trajectory [37,38]. However, there are no scales predominantly designed for the construct of PI for male nursing students. Knowledge of PI with its background variables remains limited. Therefore, improving our knowledge of the perspectives of male nursing students vis-à-vis PI is the first step in devising strategies to decrease male nursing students’ attrition [23,39].

Identity can be viewed as an autobiographical narrative, its formation and definition could be referred to some in-depth qualitative research [40,41,42]. In the present research, referring to Cruess et al.’s definition of PI for medical profession, “a representation of self, achieved in stages over time during which the characteristics, values, and norms of the medical profession are internalized, resulting in an individual thinking, acting, and feeling like a physician” [43]. We decided to develop a PI scale based on cognitive, emotional, and behavioural facets with a sample of male nursing students, and to help nursing educators understand more background variables that may be related to PI. We also took a sample of female nursing students as a contrast group to find out more about the PI of their male counterparts.

## 2. Materials and Methods

A cross-sectional survey was conducted in Taiwan which gathered data on gender, age, relevant background variables, and included the PI scale. The study was approved by the Institutional Review Board (IRB No. 202110-E101).

### 2.1. Procedures

Drawing on the literature and definitions of PI, a draft with 20 items was developed by the authors, including 8, 6, and 6 items for cognitive, emotional, and behavioural identity, respectively. Five subject matter experts were invited to scrutinise the content, in order to test the content and construct validity of the measurement tool and determine whether the conceptual clarity and question meanings were consistent across subscales [44,45]. Nursing students in Taiwan were recruited, and their privacies and rights were taken care of based on ethical protection principles during the data collection process. Their signed agreements indicating consent were obtained before administering the survey. The data of male nursing students were subjected to item selection through the item response theory using Rasch model with Conquest 2.0 software [46] to obtain a formal version of the PI scale. The data of female and male nursing students were compared in a multiple latent regression on background variables.

### 2.2. Participants

A total of 384 male nursing students from 6 out of 18 nursing schools in Taiwan were purposively recruited for this study. They constituted the main sample for the development of the PI scale with validity and reliability. Their ages ranged from 18.1 to 23.5 years. The mean age was 21.0 years, and the standard deviation for age was 0.89. A sample of 402 female nursing students were conveniently sampled from a nursing school in Taiwan. Their ages ranged from 20.0 to 24.0 years. The mean age of the female nursing students was 20.61 years, with a standard deviation of 1.36. The representative of the samples was reasonably discussed for its larger ratio in the male-nursing-student population and consistent background distributions between the male and female samples [44,45]. All participants had finished their practicum at the hospital. Table 1 presents all the other background information. Gender had no significant association with religious belief, entrance test status, parents having medical or nursing-related jobs, and interest in nursing. However, gender had a significant association with experience offering care for family members in the hospital, and suitability for the nursing profession.

### 2.3. Instrument

We drew upon the related literature and definitions of PI [8,38,43,47,48,49] to draw up the PI scale in order to measure the extent of involvement in and endorsement of ideas and values of the nursing profession among male nursing students. It had 3 factors and 20 items, that is, 8, 6, and 6 items addressing cognitive, emotional, and behavioural identity, respectively. Cognitive identity was defined as male nursing students being able to both internalise nursing professionality and identify with its professional ideas and values. Emotional identity was defined as male nursing students having a sense of duty and enthusiasm towards the nursing profession. Behavioural identity was defined as male nursing students having the willingness to engage in nursing as a professional job, and espousing professional responsibility after graduation.

The scale was self-reported and adopted Likert’s five-point scoring mechanism that ranged from 1 to 5, where 1 = strongly disagree and 5 = strongly agree. The sum of individual item scores was the scale score. The higher the scale score, the higher the PI. The content validity index of 0.93 was calculated following five experts’ evaluation, and showed a high consistent agreement vis-à-vis the item content.

### 2.4. Statistical Analysis

Statistical analysis was conducted using the Rasch model of item response theory with the partial credit model [50], which estimates one difficulty parameter and k-2 step parameters for each item. ‘k’ represents the number of scoring points. The difficulty parameter represents how difficult the participants choose the higher scoring points in general, and step parameters can be viewed as a kind of difficulty threshold from a lower scoring point to a higher scoring point. These parameter estimates were tested with unweighted and weighted MNSQ (mean square) fit indices. Both have standardised t-value types. Unweighted MNSQ is sensitive to responses to items with difficulty or step parameters far from the participants, whereas the weighted one is sensitive to responses to items with difficulty or step parameters targeted at the participants. The expected value of MNSQ is 1.0, and values ranging from 0.5 to 1.5 are reasonable and acceptable. Values that are too low imply that the scale is less productive for measurement, and values that are too high imply a distortion or degradation of the measurement system [51,52]. The unit of the parameter estimates is logit. Values can range from negative to positive infinity. For each subscale, the last item must be constrained and not be estimated for the purpose of statistical identification, which makes the sum of difficulty parameter estimates 0. Similarly, for five-point Likert-type scoring, each item has four steps but only three first step parameters need to be estimated, which makes the sum of step parameter estimates 0.

## 3. Results

Table 2 shows the estimates of difficulty and step parameter with fit indices for the 20 original items. Asterisks next to the three estimates of the difficulty parameter (items 8, 14, and 20) indicate that they are constrained, and thus, not estimated with fit indices. All MNSQ values showed reasonable outcomes close to 1.0. Therefore, we paid greater attention to *t*-value, which is close to z-value in the larger sample. Based on the t-values of difficulty parameter estimates, we set up a cut-off point of 2.0, so items 7, 15, and 17 were deleted in the first round, and item 9 was deleted in the second round. For step parameter estimates, we set up larger t-values (larger than 2.6) for the sake of avoiding the inflation of type I errors. Therefore, items 2, 4, and 10 were deleted in the third round for their unreasonable step parameter estimates. Finally, item 8, ‘I am concerned about the future development of the nursing profession’ was deleted because it seemed to involve emotional rather than cognitive identity. As a result, 12 items were retained as part of the formal PI scale.

Table 3 shows the items selected out of the total list of 12, and the content and reliability coefficients of the PI scale. We presented the means, standard deviation, and factor loadings of confirmatory factor analysis for general use. The model fit indices for the model of the 3 factors with 12 items were *χ^2^* = 68.25, *df* = 51, *p* = 0.054 (>0.01), *χ^2^*/*df* = 1.34 (<5), CFI = 0.99 (>0.90), NNFI = 0.99 (>0.90), AGFI = 0.96 (>0.90), SRMR = 0.031 (<0.06), and RMSEA = 0.030 (<0.08). All of them indicate good model fit outcomes. The correlation coefficients among the three factors ranged from 0.77 to 0.89; among the three subscales, they ranged from 0.60 to 0.65.

Table 4 presents the multiple latent regression of the PI scale and its three factors on gender and background variables. Gender had significant differences in the PI scale and its three factors. Male nursing students had higher cognitive (b = 0.353, *p* < 0.001), emotional (b = 0.298, *p* < 0.001), and behavioural identity (b = 0.415, *p* < 0.001), and scored higher on the total scale (b = 0.333, *p* < 0.001). Age had a significant negative association with behavioural identity (b = −0.103, *p* < 0.01). Having a religious belief had significant lower emotional identity (b = −0.177, *p* < 0.05) and behavioural identity (b = −0.161, *p* < 0.05) than having no religious belief. Having experiences of offering care for family members in hospital had significant higher cognitive identity (b = 0.244, *p* < 0.01) and the total scale (b = 0.146, *p* < 0.05). The nursing students who reported that they were suitable for the nursing profession had significantly lower scores on the PI total scale and its three factors (b = −0.323, −0.262, −0.350, and −0.428, *ps* < 0.05). Having interest in nursing had significantly higher scores on the PI total scale and its three factors (b = 0.373, 0.426, 0.452, and 0.310, *ps* < 0.01).

Table 5 shows multiple latent regression of the PI scale and its three factors on background variables for each gender. Age had a significant negative association with behavioural identity for male (b = −0.137, *p* < 0.05) and female (b = −0.083, *p* < 0.01) nurses. Religious belief had no significant association with the PI scale and its three factors among male nurses. However, for female nurses, having a religious belief had a significant association with lower emotional and behavioural identity, and scores on the PI total scale. Having a mother with medical or nursing-related jobs had significantly higher cognitive identity (b = 0.387, *p* < 0.05) for the males. Having caring experiences had significantly higher cognitive (b = 0.219, *p* < 0.05) and behavioural identity (b = 0.221, *p* < 0.05) for the females. The male nurses who were suitable for the nursing profession had no significant association with the PI scale and its three factors. However, for female nurses, all associations were negatively significant (b = −0.372, −0.462, −0.448, and −0.400, *ps* < 0.01). The male nurses who were interested in nursing had a significantly higher cognitive (b = 0.578, *p* < 0.01) and emotional identity (b = 0.394, *p* < 0.05), and scores on the PI total scale (b = 0.354, *p* < 0.05). The female nurses who were interested in nursing had significantly higher scores on the PI total scale and its three factors (b = 0.388, 0.266, 0.557, 0.419, *ps* < 0.05).

## 4. Discussion

Male and female nursing students had similar background distributions except for experience offering care for family members in the hospital and their suitability for the nursing profession. Male nursing students had a higher rate of experience of offering care than female ones did, and this may explain why they chose the nursing profession rather than out of academic failure, or personal/family and/or financial difficulties [7,8,9]. Although the male nursing students’ experience of offering care was not significantly associated with the total PI scale and its three factors, they were more considerate than most men. However, they had many disadvantages owing to the environment and situation [2,3,4,8,33]. Thus, most male nursing students stated that they were not suitable for the nursing profession. Their reports of less suitability for the nursing profession was not associated with the total PI scale and its three factors; and they still had a higher total PI scale and its three factors than the females.

Though the PI scale was developed using a sample of male nursing students, it was suitable for female nursing students too, because the confirmatory factor analysis showed good model fit outcomes for them: *χ^2^* = 72.97, *df* = 51, *p* = 0.023, *χ^2^*/*df* = 1.43, CFI = 0.99, NNFI = 0.99, AGFI = 0.95, SRMR = 0.033, and RMSEA = 0.030. The factor loadings for female nursing students ranged from 0.52 to 0.65, with a mean of 0.61. Thus, the PI scale can be used for both genders to compare and contrast results in multiple latent regression analyses. Multiple latent analyses can be simultaneously executed for the three factors, and can enhance the comparability of regression coefficients. The results offered valuable insights for the use of nursing educators, especially for those handling male nursing students.

We found that male nursing students had significantly higher total scores on the PI scale and its three factors than their female counterparts did. The results were inconsistent with those of past studies [5,6,10,20,23]. This may imply that the PI scale we developed for the male nursing students differed from those that had been developed in the past. The items on the PI scale were designed by a male nursing educator and manager, who considered the characteristics of male and female nursing students. The results seem encouraging for male nursing students to continue engaging in the nursing profession despite facing many disadvantages.

We found that a mother having a medical or nursing-related job had a significant association with cognitive identity among male but not female nursing students. This may imply that women engaged in medical and/or nursing-related jobs are more likely to make their sons gain a better understanding of PI in the nursing profession. Despite the effect not covering emotional and behavioural identity, it may still characterise the sons’ thinking rather than emotion and behaviour being influenced by their mother, which seems consistent with the study on mother’s education level not affecting the emotional intelligence of their male nursing students [44]. To promote the PI of male nursing students, the results suggest that educators of male nursing students should emphasise more rationality in the education process rather than just orders or commands without reasoning.

Religious belief had no association with PI among male nursing students, but had a significantly negative association with emotional and behavioural identity, and the score on the PI total scale for female nursing students. The female nursing students with religious beliefs had a lower ability to recognise the emotion of others [44]; they recognised less emotions of others, and thus, obtained less behavioural feedback from others, which thereby, may decrease their nursing PI from emotional and behavioural socialisation educational processes. However, male nursing students had more psychological capital in terms of optimism and resilience than the female ones did [40]. Thus, a religious belief did not have much of an influence on decreasing their nursing PI.

The entrance test was not significantly associated with nursing PI. Interest in nursing was significantly associated with nursing PI except for behavioural identity among male nursing students. Therefore, the government or individual schools should open up more opportunities for men with an interest in nursing so they can enter the profession, and encourage them by focusing more on behavioural PI in order to decrease entry barriers [10]. Age was negatively associated with behavioural identity and not with cognitive and emotional identity for both genders. This may imply that finding ways to maintain their behavioural identity after their graduation from school was a key point in solving the challenge of attrition among male nurses within four years of starting their careers [1,18,19,34,35,36].

## 5. Conclusions

This study relied on advanced statistical techniques to develop a PI scale and analyse and compare data from male and female nursing students. Male nursing students had higher nursing PI than female ones did in terms of their cognitive, emotional, and behavioural identity, and scores on the total scale. Some related background variables associated with nursing PI can constitute novel findings to help design professional socialisation in the education curriculum. Having mothers with medical or nursing-related jobs may help promote the cognitive PI of male nursing students. Experience of offering care for family members in hospital can help promote the PI of female nursing students but not that of male ones. Future research should focus on the goal of decreasing loss in behavioural PI for both genders after graduation and reinforcing the association between interest in nursing and behavioural PI among male nursing students. The limitations of the present research are that some connections with a larger theory of socialisation involving nursing PI were not explored and our inferences could not be proven through experimental designs; therefore, some further international comparing studies on the larger groups are necessary to confirm our findings.

## Figures and Tables

**Table 1 healthcare-10-01317-t001:** Background Information on the Nursing Students (*n* = 786).

Background Variables	*n* (%)		*χ^2^*
	Female	Male	
Religious belief			0.48
no	153 (38.1)	137 (35.7)	
yes	249 (61.9)	247 (64.3)	
Entrance test			0.04
no	367 (91.3)	352 (91.7)	
yes	35 (8.7)	32 (8.3)	
Father had a medical or nursing-related job			1.02
no	335 (83.3)	330 (85.9)	
yes	67 (16.7)	54 (14.1)	
Mother had a medical or nursing-related job			2.49
no	323 (80.3)	325 (84.6)	
yes	79 (19.7)	59 (15.4)	
Experience offering care for family members in hospital			5.71 *
no	177 (44.0)	137 (35.7)	
yes	225 (56.0)	247 (64.3)	
Suitability for the nursing profession			9.30 **
no	347 (86.3)	357 (93.0)	
yes	55 (13.7)	27 (7.0)	
Interest in nursing			0.08
no	53 (13.2)	48 (12.5)	
yes	349 (86.8)	336 (87.5)	

* *p* < 0.05. ** *p* < 0.01.

**Table 2 healthcare-10-01317-t002:** The parameter estimates of item response theory with partial credit model for the 20 original items on the PI scale (*n* = 384).

				Unweighted Fit		Weighted Fit						Unweighted Fit		Weighted Fit	
Item	Difficulty Estimate		Error	MNSQ	*t*	MNSQ	*t*	Item	Difficulty Estimate		Error	MNSQ	*t*	MNSQ	*t*
1	0.340		0.045	1.02	0.2	1.05	0.7	11	−0.002		0.046	0.87	−1.8	0.89	−1.5
2	0.145		0.047	0.91	−1.3	0.92	−1.1	12	0.121		0.045	0.95	−0.7	1.01	0.1
3	0.055		0.046	1.00	−0.0	1.06	0.8	13	0.118		0.045	0.87	−1.9	0.91	−1.2
4	−0.066		0.047	0.94	−0.9	0.99	−0.0	14	−0.076 *						
5	−0.240		0.047	0.99	−0.1	1.08	1.0	15	−0.015		0.045	1.12	1.7	1.18	2.3
6	−0.120		0.046	0.88	−1.7	0.97	−0.4	16	0.070		0.045	1.01	0.1	1.10	1.4
7	−0.147		0.047	0.83	−2.5	0.90	−1.4	17	0.078		0.045	1.09	1.2	1.16	2.0
8	0.034 *							18	−0.008		0.045	0.95	−0.7	1.04	0.6
9	0.005		0.046	1.03	0.4	1.09	1.2	19	−0.185		0.046	0.98	−0.3	1.04	0.6
10	−0.166		0.047	1.01	0.2	1.00	0.1	20	0.061 *						
				**Unweighted Fit**		**Weighted** **Fit**						**Unweighted Fit**		**Weighted** **Fit**	
**Item**	**Step**	**Estimate**	**Error**	**MNSQ**	** *t* **	**MNSQ**	** *t* **	**Item**	**Step**	**Estimate**	**Error**	**MNSQ**	** *t* **	**MNSQ**	** *t* **
1	1	−0.872	0.123	0.96	−0.5	0.98	−0.4	11	1	−1.911	0.130	1.01	0.1	0.97	−0.3
1	2	−1.277	0.113	0.94	−0.8	0.97	−0.7	11	2	−0.438	0.112	0.95	−0.7	0.96	−0.9
1	3	0.612	0.117	0.95	−0.6	0.97	−0.8	11	3	0.341	0.109	0.87	−1.9	0.91	−2.9
2	1	−1.313	0.130	1.03	0.5	0.98	−0.3	12	1	−1.057	0.126	1.04	0.6	1.02	0.3
2	2	−1.117	0.116	0.94	−0.9	0.96	−0.9	12	2	−1.157	0.114	0.95	−0.6	0.97	−0.6
2	3	0.363	0.111	0.88	−1.7	0.93	−2.4	12	3	0.423	0.110	0.93	−1.0	0.95	−1.5
3	1	−1.406	0.125	1.10	1.3	1.03	0.6	13	1	−0.829	0.121	0.97	−0.5	0.95	−0.8
3	2	−0.868	0.113	0.94	−0.9	0.95	−1.1	13	2	−1.012	0.112	0.93	−1.0	0.95	−1.3
3	3	0.487	0.113	0.86	−1.9	0.93	−1.9	13	3	0.132	0.109	0.92	−1.0	0.95	−1.7
4	1	−1.778	0.130	1.02	0.3	0.96	−0.6	14	1	−0.968	0.121	1.04	0.5	1.02	0.4
4	2	−0.807	0.114	0.91	−1.3	0.94	−1.4	14	2	−1.110	0.112	0.96	−0.5	0.98	−0.4
4	3	0.510	0.111	0.88	−1.8	0.93	−2.2	14	3	0.430	0.110	0.90	−1.4	0.95	−1.6
5	1	−1.672	0.124	1.05	0.8	0.99	−0.1	15	1	−1.396	0.122	0.98	−0.3	0.98	−0.3
5	2	−0.543	0.112	0.95	−0.6	0.97	−0.7	15	2	−0.693	0.111	0.98	−0.2	0.99	−0.1
5	3	0.164	0.109	0.92	−1.1	0.96	−1.5	15	3	0.336	0.109	0.96	−0.6	0.96	−1.3
6	1	−1.101	0.121	1.04	0.6	1.04	0.7	16	1	−1.463	0.127	1.17	2.3	1.06	1.0
6	2	−0.980	0.112	0.97	−0.4	1.00	−0.0	16	2	−0.888	0.114	1.02	0.3	1.02	0.4
6	3	0.451	0.112	0.89	−1.5	0.95	−1.3	16	3	0.414	0.108	0.97	−0.4	0.98	−0.7
7	1	−1.809	0.128	0.99	−0.2	0.99	−0.2	17	1	−1.282	0.123	1.09	1.2	1.02	0.3
7	2	−0.759	0.113	0.96	−0.5	0.98	−0.5	17	2	−0.730	0.111	0.95	−0.7	0.96	−1.1
7	3	0.575	0.111	0.99	−0.1	0.97	−0.8	17	3	0.238	0.108	0.90	−1.5	0.94	−2.2
8	1	−1.640	0.128	1.09	1.3	0.99	−0.1	18	1	−1.814	0.128	1.07	1.0	1.05	0.8
8	2	−0.748	0.113	0.93	−0.9	0.95	−1.1	18	2	−0.652	0.112	1.03	0.4	1.03	0.8
8	3	0.397	0.111	0.92	−1.2	0.95	−1.6	18	3	0.544	0.109	0.93	−1.0	0.97	−1.1
9	1	−1.270	0.127	0.95	−0.6	0.96	−0.5	19	1	−1.918	0.127	1.19	2.5	1.05	0.8
9	2	−1.112	0.115	0.94	−0.8	0.95	−0.9	19	2	−0.541	0.113	0.99	−0.1	1.00	−0.0
9	3	0.492	0.109	0.97	−0.4	0.99	−0.4	19	3	0.315	0.107	0.93	−0.9	0.96	−1.8
10	1	−1.962	0.133	1.27	3.5	1.12	1.7	20	1	−1.019	0.119	1.21	2.7	1.14	2.5
10	2	−0.697	0.115	1.00	−0.0	1.01	0.3	20	2	−0.849	0.110	1.08	1.1	1.08	2.3
10	3	0.422	0.107	0.94	−0.8	0.95	−1.7	20	3	0.517	0.113	1.00	−0.0	1.00	−0.0

Note: The top panel of the table represents the difficulty parameter estimates. The bottom panel represents step parameter estimates. Asterisks * next to the difficulty parameter estimates indicate that they are constrained. Items 1 to 8 belong to cognitive identity. Items 9 to 14 belong to emotional identity. Items 15 to 20 belong to behavioural identity.

**Table 3 healthcare-10-01317-t003:** Factor loadings and reliability coefficients of the PI scale (*n* = 384).

Items	Mean (Standard Deviation)	Factor Loading	Cronbach’s Alpha
1. I generally understand the connotations of nursing professionality	3.58 (1.16)	0.71	
2. I am full of confidence in nursing prosperity	3.73 (1.08)	0.73	
3. I know that the nursing profession has requirements of professional competence of students	3.89 (1.01)	0.67	
4. I have a positive evaluation of the nursing profession	3.87 (1.07)	0.68	
Cognitive identity	15.07 (3.37)		0.79
5. I enjoy the processes of learning the art of nursing	3.71 (1.03)	0.69	
6. I consider the job of a nurse pleasurable	3.72 (1.04)	0.63	
7. I am proud of being a nursing student	3.77 (1.06)	0.67	
8. Becoming a nurse will make my life values come true	3.86 (1.01)	0.66	
Emotional identity	15.06 (3.15)		0.76
9. After graduation, I will choose to work as a nurse	3.79 (1.00)	0.63	
10. After graduation, I will still look for resources to promote nursing professionality	3.79 (1.00)	0.71	
11. Working as a nurse can help me achieve self-realisation	3.89 (0.94)	0.61	
12. My professional performance helps me satisfy the requirements of working as a nurse	3.86 (1.09)	0.56	
Behavioural identity	15.33 (2.97)		0.72
Total scale	45.46 (8.22)		0.88

Note: ‘Cognitive identity’ includes the original 1st, 3rd, 5th, and 6th items in Table 1. ‘Emotional identity’ includes the 11th−14th items. ‘Behavioural identity’ includes the 16th and 18th−20th items.

**Table 4 healthcare-10-01317-t004:** The multiple latent regression of the PI scale and its three factors on gender and background variables (*n* = 786).

Variables	Cognitive Identity	Emotional Identity	Behavioural Identity	Total PI Scale
Constant	0.678 (0.723)	1.191 (0.691)	2.782 *** (0.729)	1.464 * (0.648)
Gender	0.353 *** (0.075)	0.298 *** (0.072)	0.415 *** (0.076)	0.333 *** (0.067)
Age	−0.029 (0.032)	−0.042 (0.031)	−0.103 ** (0.032)	−0.055 (0.029)
Religious belief	−0.041 (0.077)	−0.177 * (0.074)	−0.161 * (0.078)	−0.117 (0.069)
Entrance test	−0.054 (0.132)	−0.197 (0.126)	−0.221 (0.133)	−0.152 (0.118)
Father having a related job	−0.174 (0.111)	0.109 (0.106)	−0.112 (0.112)	−0.053 (0.100)
Mother having a related job	0.141 (0.106)	0.074 (0.101)	0.113 (0.107)	0.100 (0.095)
Experience offering care in a hospital	0.244 ** (0.076)	0.091 (0.073)	0.140 (0.077)	0.146 * (0.068)
Suitability for the nursing profession	−0.262 * (0.126)	−0.350 ** (0.121)	−0.428 *** (0.127)	−0.323 ** (0.113)
Interest in nursing	0.426 *** (0.113)	0.452 *** (0.108)	0.310 ** (0.114)	0.373 *** (0.102)

Note: * *p* < 0.05, ** *p* < 0.01, *** *p* < 0.001. The latent regression coefficients in cells with standard errors are in parentheses. All background variables except for gender and age were binary (no and yes) variables. The female is a reference group for gender, and ‘no’ is a reference group for other variables except for age.

**Table 5 healthcare-10-01317-t005:** Multiple latent regression of the PI scale and its three factors on background variables for each gender.

Variables	Cognitive Identity	Emotional Identity	Behavioural Identity	Total PI Scale
Male nursing students				
Constant	0.496 (1.549)	0.786 (1.310)	3.915 ** (1.422)	1.653 (1.247)
Age	−0.027 (0.070)	−0.028 (0.059)	−0.137 * (0.064)	−0.062 (0.057)
Religious belief	0.062 (0.130)	−0.022 (0.110)	−0.115 (0.119)	−0.021 (0.105)
Entrance test	−0.121 (0.224)	−0.342 (0.190)	−0.255 (0.206)	−0.228 (0.181)
Father having a related job	−0.226 (0.198)	0.132 (0.168)	−0.142 (0.182)	−0.069 (0.160)
Mother having a related job	0.387 * (0.192)	0.204 (0.162)	0.312 (0.176)	0.271 (0.154)
Experience offering care in a hospital	0.251 (0.131)	0.101 (0.111)	0.040 (0.121)	0.118 (0.106)
Suitability for the nursing profession	−0.131 (0.248)	−0.273 (0.210)	−0.319 (0.228)	−0.214 (0.200)
Interest in nursing	0.578 ** (0.189)	0.394 * (0.160)	0.194 (0.174)	0.354 * (0.152)
**Female nursing students**				
Constant	1.273 (0.724)	2.347 * (0.954)	2.284 ** (0.745)	1.843 * (0.737)
Age	−0.034 (0.031)	−0.066 (0.041)	−0.083 ** (0.032)	−0.058 (0.032)
Religious belief	−0.132 (0.087)	−0.374 ** (0.115)	−0.199 * (0.090)	−0.215 ** (0.089)
Entrance test	−0.013 (0.149)	−0.072 (0.196)	−0.166 (0.153)	−0.078 (0.151)
Father having a related job	−0.163 (0.121)	0.029 (0.159)	−0.088 (0.124)	−0.077 (0.123)
Mother having a related job	−0.037 (0.114)	−0.050 (0.149)	−0.024 (0.117)	−0.033 (0.116)
Experience offering care in a hospital	0.219 * (0.086)	0.070 (0.113)	0.221 * (0.089)	0.167 (0.088)
Suitability for the nursing profession	−0.372 ** (0.130)	−0.462 ** (0.171)	−0.448 *** (0.133)	−0.400 ** (0.132)
Interest in nursing	0.266 * (0.131)	0.557 ** (0.172)	0.419 ** (0.134)	0.388 ** (0.133)

Note: * *p* < 0.05, ** *p* < 0.01, *** *p* < 0.001. The regression coefficients in cells with standard errors are in parentheses. All background variables except for age were binary variables (no and yes). The ‘no’ is a reference group for each background variable except for age.

## Data Availability

The data presented in this study are available upon reasonable request.

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
