# Peer review of "Professional Identity Scale for Male Nursing Students Using the Rasch Model and Latent Regression on Gender and Background Variables"

_healthcare, 2022, doi:10.3390/healthcare10071317_

Round 1

Reviewer 1 Report

1.  Introduction: The research topic is necessary for the actual educational field and has the potential to contribute to the nursing knowledge body. The contents of the main concepts are logically well organized.

2. Materials and Methods

2.1. Procedures: Please describe the interventions attempted for ethical protection of the subject during the data collection process.

2.2. Participants: Did you use convenience sampling among non-probability sampling as the sampling method? What is the basis for calculating the number of samples?

4. Discussion : Based on the research results, the opinions of previous studies and researchers are logically described.

We hope to improve the completeness of the thesis by making only a few corrections as advised above.

Author Response

Dear Editor:

Reviewer 2 Report

Thank you for the opportunity to review this manuscript. The time spent on its creation and submission is greatly appreciated. The topic you raise is interesting. There is a widespread problem regarding the number of nurses globally. Progress in this field is important. Here are my recommendations, I hope they are useful:

-The introduction needs to expose the definitions and models of professional identity from which it starts. To present the creation of a questionnaire, it is necessary to present the initial theoretical model from which it starts. This is extremely important for your work. How did you decide the factors into which you were going to group the items? Based on what did they raise those items? What is the initial model from which to propose the instrument and its factors? This needs a higher degree of justification both in the introduction and in the method. 

-Was an exploratory factorial analysis carried out to confirm the grouping of items as proposed?

-Were all female participants chosen from the same school? Justify why they were not selected from the same 4 schools as the male participants.

-As you indicate, the scale was suitable for female nurses. But the title of the manuscript and the scale imply its validity only for the male gender. Please modify this.

-Include the limitations of your study.

Author Response

Dear Editor

Reviewer 3 Report

The article is generally very well written and almost everything is well presented.

I have only 3 suggestions:

1. improving citations- I suggest to write a little bit more (even one paragraph) about problems with male identity in the modern world and about concepts of identity at all (another paragraph)

this articles could be helpful as touching concepts of identity:

Petrovska, I. (2021). Psychological model of civic identity formation. Journal of Education Culture and Society, 12(2), 167–178. https://doi.org/10.15503/jecs2021.2.167.178

Dan, M. . (2020). Early childhood identity: ethnicity and acculturation . Journal of Education Culture and Society, 5(1), 145–157. https://doi.org/10.15503/jecs20141.145.157

Jakubowska, L. . (2020). Identity as a narrative of autobiography. Journal of Education Culture and Society, 1(2), 51–66. https://doi.org/10.15503/jecs20102.51.66

2. writing about limitations of the study- I suggest to write about necessity of further international comparing studies on the larger groups. Limitations of the study are usually set up after conclusions

3. adding conclusions. The conclusions are very short because are not connected with larger theory of socialization.  Writing about problems with identity and comparing with achieved results could make the scientific work more complex

Author Response

Dear Editor:

Round 2

Reviewer 2 Report

Thank you for attending and responding to my comments. Good work!